# Invasive Pneumococcal Disease in People Living with HIV: A Retrospective Case—Control Study in Brazil

**DOI:** 10.3390/tropicalmed8060328

**Published:** 2023-06-19

**Authors:** Roxana Flores Mamani, Tiago de Assunção López, Waldir Madany Jalo, Marcelo Ribeiro Alves, Estevão Portela Nunes, Mario Sérgio Pereira, Erica Aparecida dos Santos Ribeiro da Silva, Maria Cristina da Silva Lourenço, Valdiléa Gonçalves Veloso, Beatriz Jegerhorn Grinsztejn, Sandra Wagner Cardoso, Cristiane da Cruz Lamas

**Affiliations:** 1Instituto Nacional de Infectologia Evandro Chagas, Fiocruz, Avenida Brasil 4365-Manguinhos, Rio de Janeiro 21040-360, RJ, Brazil; waldirjalo@hotmail.com (W.M.J.); marcelo.ribeiro@ini.fiocruz.br (M.R.A.); estevao.nunes@ini.fiocruz.br (E.P.N.); msergiopereira@gmail.com (M.S.P.); erica.aparecida@ini.fiocruz.br (E.A.d.S.R.d.S.); cristina.lourenco@ini.fiocruz.br (M.C.d.S.L.); valdilea.veloso@ini.fiocruz.br (V.G.V.); beatriz.grinsztejn@gmail.com (B.J.G.); dra.wagner@gmail.com (S.W.C.); cristianelamas@gmail.com (C.d.C.L.); 2Barra da Tijuca Campus, Department of Medicine, Universidade do Grande Rio/Afya, Avenida Ayrton Senna, 2.200, Barra da Tijuca 22775-003, RJ, Brazil; tiagodeassuncaolopez@gmail.com; 3Instituto Nacional de Cardiologia, Rua das Laranjeiras, 374-Laranjeiras, Rio de Janeiro 22240-006, RJ, Brazil

**Keywords:** HIV/AIDS, *Streptococcus pneumoniae*, pneumococcus, invasive pneumococcal disease, vaccination, Brazil

## Abstract

HIV-infected patients are at particular risk for invasive pneumococcal disease (IPD). We describe cases of IPD in people living with HIV/AIDS (PLWHA) and find associated risk factors for infection and death. Methods: A retrospective case-control study, nested in a cohort, including PLWHA with and without IPD, conducted in Brazil, 2005–2020. Controls were of the same gender/age and seen at the same time/place as cases. Results: We identified 55 episodes of IPD (cases) in 45 patients and 108 controls. The incidence of IPD was 964/100,000 person-years. A total of 42 of 55 (76.4%) IPD episodes presented with pneumonia and 11 (20%) with bacteremia without a focus and 38/45 (84.4%) were hospitalized. Blood cultures were positive in 54/55 (98.2%). Liver cirrhosis and COPD were the only factors associated with IPD in PLWHA in univariate analysis, although no associated factors were found in multivariate analysis. Penicillin resistance was found in 4/45 (8.9%). Regarding antiretroviral therapy (ART), 40/45 (88.9%) cases vs. 80/102 controls (74.1%) were in use (*p* = 0.07). Patients with HIV and IPD had a higher CD4 count of 267 cells/mm^3^ compared with the control group, in which it was 140 cells/mm^3^ (*p* = 0.027). Pneumococcal vaccination was documented in 19%. Alcoholism (*p* = 0.018), hepatic cirrhosis (*p* = 0.003), and lower nadir CD4 count (*p* = 0.033) were associated with the risk of death in patients with IPD. In-hospital mortality among PLWHA and IPD was 21.1%, and it was associated with thrombocytopenia and hypoalbuminemia, elevated band forms, creatinine, and aspartate aminotransferase (AST). Conclusions: The incidence of IPD in PLWHA remained high despite ART. The vaccination rate was low. Liver cirrhosis was associated with IPD and death.

## 1. Introduction

Invasive pneumococcal disease (IPD) is an infection confirmed by isolating *S. pneumoniae* from sterile sites. *S. pneumoniae* is the leading etiologic agent of community-acquired pneumonia, meningitis, sinusitis, and otitis media. In bacteremia, secondary complications such as endocarditis, arthritis, or meningitis may occur [1].

*S. pneumoniae* is a Gram-positive encapsulated coccus with 100 immunologically distinct serotypes [2]. It is one of the most important bacterial pathogens in people living with HIV/AIDS (PLWHA). Rates of pneumonia are 25 times more frequent among these patients [3]. The incidence of IPD in this population is about 46 to 100 times higher than in the general population, and the risk is particularly high in those with CD4 count < 200/mm^3^ [1]. Several studies show that the incidence of IPD decreases with combined antiretroviral therapy (ART), but it remains significantly higher than that of the general population [4,5,6]. Patients who develop a pneumococcal infection in the context of immunosuppression are more likely to develop bacteremia with no evident focus and septic shock, and their risk of death is almost three times higher compared with non-immunosuppressed patients [7].

Several risk factors associated with IPD in PLWHA have been identified: age > 60 years and aboriginal ethnicity [8], education less than high school [8], smoking and alcoholism [4,9], intravenous drug use (IDU) [4,8,10], hepatitis C co-infection [8], chronic obstructive pulmonary disease (COPD) [8], low serum albumin, CD4 < 200 cells/mm^3^ [8], CD4 < 100 cells/mm^3^ [4,9], HIV RNA > 50.000 copies/mL, no use of ART [4], not having received 23-valent pneumococcal vaccine (PCV23) [4], and prior hospitalization [4,9]. Protective factors for IPD were ART [9] and pneumococcal vaccination [8,9,10].

This study may contribute to the global understanding of invasive pneumococcal disease in patients with HIV since it is a disease that affects all human beings, regardless of geographical location, and is associated with high rates of morbidity and mortality, particularly in developing countries. The meta-analysis showed that there was a higher incidence of IPD in African countries compared with non-African countries [11]. Brazil is a middle-income country located in South America. The Brazilian scenario, therefore, may apply to other Latin American countries and other middle-income countries.

This study aimed (i) to evaluate the incidence of IPD among PLWHA seen at the Instituto Nacional de Infectologia Evandro Chagas/Fundação Oswaldo Cruz (INI/Fiocruz), Rio de Janeiro, Brazil, and (ii) to describe the profile of affected patients, including treatment and outcomes, and (iii) to find risk factors associated with this disease in PLWHA.

## 2. Materials and Methods

This is a retrospective case-control (1:2) study, nested in a cohort, including PLWHA with and without IPD, from January 2005 to June 2020. HIV care and treatment have been provided at INI/Fiocruz since 1986. A longitudinal observational clinical cohort of PLWHA receiving care at INI has been maintained by the Clinical Research Laboratory on HIV/AIDS (LAPCLIN/AIDS). Cohort data are updated regularly and have contributed to several studies [12]. From 1986 to 2019, INI’s comprehensive observational database (HIV cohort) maintained information on 7185 PLWHA ≥ 18 years old. We estimate around 5000 people in active follow-up (i.e., those who attended INI at least once in the last year for lab evaluation, medication refill collection, or consultation).

Data Collection: Microbiological data were collected from the results made available by the Laboratory of Bacteriology and Bioassays in an automated system and from written internal laboratory records. Study variables were obtained from the HIV cohort database and complemented by data extracted from the electronic medical records. The isolation method was performed by direct inoculation of clinical samples in enriched culture media (chocolate agar and blood agar) and incubation at 5% CO_2_, at a temperature of 35 ± 1 °C, within a maximum of 72 h after receiving the samples. If samples were blood and/or sterile fluid samples, they were inoculated into automated culture flasks (Bactec/Becton Dickson). Incubation time was up to 5 days. When positive, blood and sterile fluid were inoculated in enriched media, as previously described. The antimicrobial susceptibility test was performed by the agar diffusion method (disk diffusion), and the determination of the minimum inhibitory concentration by gradient strip agar diffusion was made. We followed the interpretative criteria of the CLSI—Clinical and Laboratory Standards Institute. Molecular tests and serotyping were not performed.

Study population: Adult PLWHA with IPD (cases) and PLWHA without IPD (controls). Controls were patients of the same sex at birth and age group (±5 years), seen in the same semester as the cases, in the same setting (medical clinic/emergency room or medical ward/medical intensive care unit).

Study variables: Sex at birth, age at the time of the IPD episode, CD4+ T cell count, CD8+ T cell count, CD4/CD8 ratio, and viral load (all obtained within 6months before IPD episode), use of ART, prophylaxis with cotrimoxazole, pneumococcal vaccination, comorbidities, such as diabetes mellitus, COPD, liver cirrhosis, chronic kidney failure, use of chemotherapy, steroids, inhaled or intravenous cocaine use, crack use, alcohol abuse, smoking, whether the patient was hospitalized or not, length of hospital and intensive care unit (ICU) stay and procedures performed (mechanical ventilation, use of vasopressors, hemodialysis), antibiotic use (drug name), samples collected for culture, the sensitivity of the isolate to penicillin and cotrimoxazole, case outcome (death or not), and need for additional therapeutic procedures.

Definitions: IPD was defined as the isolation of *S. pneumoniae* from at least one blood culture, from cerebrospinal fluid (CSF) culture, or other sterile sites. Prophylaxis with sulfamethoxazole/trimethoprim was considered as the use of the drug for more than 2 weeks and up to 3 months before isolation of *S. pneumoniae*. Community-acquired infection was considered if the patients remained as outpatients and had no previous recent hospital admissions, or if admitted, the diagnosis occurred within 48 h after admission. Hospital-acquired infection was diagnosed if the event occurred more than 48 h after admission. Pneumonia was defined as an acute or subacute pulmonary condition characterized by the following: fever and/or cough; new purulent sputum; dyspnea or tachypnea and pleuritic chest pain, with or without radiological confirmation, and with adequate antibiotic treatment administered [13], as well as isolation of pneumococcus in a sterile sample. Bacteremia was defined as the isolation of *S. pneumoniae* from peripheral blood cultures. Meningitis was defined as the isolation of *S. pneumoniae* in cerebral spinal fluid associated with headache, fever, altered mental state, and/or neck stiffness. The use of ART was defined by its prescription at least one month before the event. Nadir T-CD4+ count was considered the lowest level ever of CD4+ count in the patient’s history, and T-CD4+ count was the CD4+ cell count measured closest to the IPD episode, up to 6 months previously. Pneumococcal vaccine prophylaxis was considered when the vaccine (any pneumococcal vaccine) was given more than 2 weeks before the episode up to 5 years before the isolation of *S. pneumoniae*. Infection was defined as recurrent when *S. pneumoniae* was isolated more than 30 days after the date of the previous event. Mortality was defined as related to the IPD when the death occurred within 30 days of *S. pneumoniae* isolation. Penicillin resistance was measured by the disc diffusion method with the oxacillin disc; the cutoff levels used were those established by The Clinical and Laboratory Standards Institute (CLSI) [14], and differentiation was made if the biological material was blood or other fluids (non-meningeal cohort point > 8 µg/mL) and CSF (meningeal cohort point > 0.12 µg/mL). Macrolide resistance was evaluated with the erythromycin disc, which was used to determine sensitivity to azithromycin and clarithromycin (cutoff point < 15 mm).

Statistical analysis: Mann–Whitney U tests were used to compare baseline demographic and clinical variables with continuous numerical variables. For categorical nominal variables, Pearson’s χ^2^ tests were used to assess frequency independence between these variables and either pneumococcal infection (cases and controls) or pneumococcal infection-related death (cases). Both cases and controls were used to estimate the risks of pneumococcal disease. The effects of various risk factors on pneumococcal disease were assessed using odds ratios (aOR) and their corresponding 95% confidence intervals (CI), estimated using multiple logistic regression models. To rule out any possible bias introduced by convenience sampling, we introduced as confounders in these models the patients’ age, sex at birth, race/color, level of education, household income, nadir T-CD4+ count, T-CD4+ count, and viral load. To estimate the risks of progression to death from pneumococcal infection (cases only), we calculated person-years (pY) by measuring the years of follow-up of at-risk patients living with HIV/AIDS and years of follow-up after their last episode of pneumococcal infection, respectively. Living subjects were censored at the end of the follow-up (30 June 2021). In addition, the incidence rate of death related to pneumococcal infection and its 95% CI were estimated according to asymptotic standard errors calculated from Gamma distributions [15]. Regardless of the model, whenever necessary, we categorized the continuous numerical variables using the round integer closest to their medians as cutoff points. Two-tailed significance levels were less than or equal to 0.01, 0.05, and 0.1. All statistical analyses were performed using R version 4.1.0 (R Core Team, 2021). The annual incidence of IPD in PLWHA was calculated by dividing the number of IPD episodes by the number of new PLWHA included in the cohort each year. To identify an increasing or decreasing trend in IPD, we used a nonparametric Spearman test between the observations and time estimated in bootstrap samples (R = 1000).

Ethical approval: The study was approved by the Ethics Committee of the Instituto Nacional de Infectologia Evandro Chagas/Fiocruz (IRB no 32449420.4.1001.5262), number 4,133,994 on 13 July 2020.Terms of free and informed consent were waived.

## 3. Results 

Among 5434 HIV patients followed in the study period, the overall incidence of IPD in PLWHA was 964 episodes/100,000 person-years between 2005 and 2020 at INI/Fiocruz.

The annual incidence of IPD in PLWHA/100,000 person-years is presented in Figure 1.

In the calendar periods 2005–2009, 2010–2014, and 2015–2019, the incidence of IPD per 100,000 person-years increased from 710 (95% CI, 270–1150) to 1050 (95% CI, 520–1580) to 1453 (95% CI, 946–1960), respectively. We found no significant differences between the years 2005 and 2019 for the mean incidences of IPD per 100,000 person-years, as evidenced by the overlap of the estimated 95% confidence intervals, but we did find a trend of a 1.4% increase per year (rho = 0.014; *p*-value < 0.001).

During the study period, 55 episodes (cases) of IPD were identified in 45 patients. A total of 17 IPD episodes were recurrent in 7 patients (1 patient had 4 episodes, 1 patient had 3, and the remaining 5 had 2 episodes each). Most cases required hospitalization (38/45, 84.4%), and 7 (15.6%) were diagnosed and treated on an outpatient basis. About two-thirds of the patients were male at birth, and the median age was 42 (35–48) years. Self-declared brown and black people accounted for about three-quarters of the sample.

Among episodes of IPD, bacterial pneumonia occurred in 42/55 (76.4%), primary bacteremia in 11 (20%), and pneumococcal meningitis in 2 (3.6%). *S. pneumoniae* was isolated in blood cultures in 54/55 (98.2%) episodes. In one episode, the bacterium was isolated in blood and CSF, and in one episode, in CSF only.

The 55 episodes of IPD in PLWHA were paired with 108 PLWHA controls without IPD.

Table 1 shows demographic and clinical variables in HIV-positive patients during their first episode of IPD compared with HIV-positive patients without IPD, stratified according to the presence or absence of invasive pneumococcal disease. Recent diagnosis of HIV infection was less frequent among patients with IPD. Overall, 120/153 (78.4%) were on ART, and the frequency of use was higher in cases compared with controls (40/45 or 88.9% vs. 80/108 or 74.1%, *p* = 0.07, respectively). Patients with HIV and IPD had a higher CD4 count of 267 cells/mm^3^ compared with the control group, in which it was 140 cells/mm^3^ (*p* = 0.027).

We had 15 patients in the control group who died (14.9%); in 8/15 (53%) the cause was respiratory (pulmonary tuberculosis, pneumocystis pneumonia, or other bacterial pneumonia), in 5/15 (33%), the cause was neurological (meningoencephalitis, neurotoxoplasmosis, or AIDS encephalitis), and in 2/15 (13%) the cause was hematological (Kaposi’s sarcoma and lymphoma).

Adjusted demographic and clinical characteristics of PLWHA with IPD who died in hospital are presented in Appendix A; only alcoholism (*p* < 0.018) and hepatic cirrhosis (*p* < 0.003) remained associated with the risk of death in patients with IPD.

Data on hospitalization and death are presented in Appendix A. In-hospital mortality (within 30 days) in our cohort was 21.1% in cases and 14.9% in controls, while mortality in the follow-up period (1 year after the end of the study, June 2021) was 42.2% in cases and 40.7% in controls, with no statistical difference between groups.

Regarding PLWHA and IPD who died, there was a significant difference in nadir (i.e., lowest ever) CD4 counts, which had a median of 53.5 [IQR 12;93.5] cells/mm^3^ compared with a median of 151.5 [80.75;334] in those who lived (*p* = 0.033). Alcoholism and liver cirrhosis were also associated with death in PLWHA and IPD (Appendix A).

In the case group, there were 8 deaths: 7/8 (88%) from pulmonary causes (severe pneumonia) and 1/8 (12%) from primary bacteremia. We had 15 patients in the control group who died (14.9%): in 8/15 (53%) the cause was a respiratory failure (pulmonary tuberculosis, pneumocystosis, or other bacterial pneumonia), in 5/15 (33%) the cause was neurological (meningoencephalitis, neurotoxoplasmosis, or AIDS encephalitis), and in 2/15 (13%), the cause was hematological (Kaposi’s sarcoma and lymphoma).

The antibiotic regimens used to treat IPD were of the penicillin class (intravenous ampicillin or amoxicillin-clavulanate) in 8/54 (14.8%), ampicillin or amoxicillin-clavulanate combined with levofloxacin or a macrolide in 23/54 (42.6%), cephalosporins in 4 (7.4%), a cephalosporin combined with levofloxacin or a macrolide in 12 (22.2%), and levofloxacin in 7 (13.0%).

Pneumococcal vaccination with the 23-valent pneumococcal vaccine (PPV23) was documented in 9/45 (20%) cases and 20/108 (18.5%) controls.

*S. pneumoniae* showed resistance to penicillin in 4/45 (8.9%) cases, to cotrimoxazole in 6/45 (13.3%), and to macrolides (erythromycin) in 4/45 (8.9%). All four cases of resistance represented blood culture isolates.

The majority of IPD episodes had a clinical respiratory presentation, corresponding to 36/45 (80%) vs. 35/108 (32.4%) in controls (*p* < 0.001). On the other hand, the neurological presentation was more frequent in controls, corresponding to 22/108 (20.4%) vs. 1/45 (2.2%) in cases. These results are presented in Appendix A. Main opportunistic infections (tuberculosis, pneumocystis pneumonia, neurotoxoplasmosis, and meningeal cryptococcosis) associated with cases and controls are shown in Appendix A.

Vital signs and laboratory data comparing cases and controls are presented in Table 2. Cases presented significantly more hypotension, tachycardia, tachypnea, fever, and hypoxemia (by pulse oximetry) than controls. There were significantly more leukocytosis, neutrophilia, band forms, and lymphopenia among cases, as well as higher C-reactive protein values (CRP). As for biochemistry, cases presented higher creatinine, urea, AST, and total bilirubin levels, while serum albumin was lower. After adjustment, the variables that remained statistically significant were as follows: mean arterial pressure < 70 mmHg, tachypnea, elevated CRP, thrombocytopenia, leukocytosis, elevated band forms and neutrophilia, lymphopenia, elevated urea, creatinine, and bilirubin, and serum albumin less than 2.3 mg/dL, as presented in Table 3. These parameters are essentially related to bacterial sepsis.

The following risk factors were associated with IPD in PLWHA: COPD (28.9% vs. 10.3%, *p* = 0.009) and liver cirrhosis (20% vs. 5.6%, *p* = 0.015) per univariate analysis; when the variables were adjusted in the risk factor analysis, no statistically significant factor was found to be associated with IPD as shown in Table 4.

## 4. Discussion

Ours was a retrospective study on IPD in PLWHA in Rio de Janeiro, Brazil, in the years 2005 to 2020. The incidence of IPD was high as was the associated in-hospital mortality in patients living with HIV in our cohort during the study period. Alcoholism and liver cirrhosis were the only factors associated with death. From 1980 to June 2020, 1,011,617 PLWHA were identified in Brazil, with an average of 39,000 new cases in the last 5 years. The annual number of PLWHA has been decreasing since 2013 when it reached 43,368 cases; in 2019, 37,308 cases were registered. The detection rate of HIV/AIDS cases has been falling in Brazil in recent years. In 2011, this rate was 22.2 cases/100,000 inhabitants; in 2015, 20.1; in 2017, it dropped to 18.6; and in 2019, it reached 17.8 cases/100,000 inhabitants. In a 10-year period, the detection rate showed a drop of 17.2% (according to data from the 2019 Epidemiological Bulletin [16]).

Regarding the prevention of IPD and pneumococcal pneumonia, in Brazil, the 10-valent pneumococcal vaccine has been introduced since 2010, through the PNI (National Immunization Program). Currently, three pneumococcal vaccines are registered for use: the 10-valent pneumococcal vaccine (PCV10), the 13-valent pneumococcal vaccine (PCV13), and the23-valent pneumococcal vaccine (PPV23). The vaccination scheme is as follows: for children under 5 years and with PCV10, 2 doses and 1 booster; for adults over 60 years, 1 dose of PPV23; and for patients living with HIV and other immunosuppressed patients, 1 dose of PCV13 followed by PPV23 6 to 12 months later, and a second dose of PPV23 5 years after the first. According to data from the Ministry of Health, the vaccination coverage of the population regarding the target audience of immunosuppressed patients was 67% in 2020 and 73% in 2019 [17].

PLWHA have a higher risk of noninvasive and invasive pneumococcal disease, which is one of this group’s most frequent opportunistic infections, as demonstrated in several studies [4,5,6,8,18]. We have studied episodes of IPD in PLWHA in a reference institute for Infectious Diseases in Rio de Janeiro, Brazil, from 2005 to 2020, and have compared demographic, clinical, and laboratory features of control patients, selected based on age, gender, and healthcare scenario. The incidence of IPD was calculated based on the cohort of PLWHA that are followed up in our institute. In our study, we included 45 patients and 55 episodes of IPD, which we matched with 108 controls. We found an incidence of 964 episodes/100,0000 person-years in the study period of 2005 to 2020; the incidence in the literature ranges from 245 to 1094 cases per 100,000 person-years in studies conducted from 1990–2015 in the United States and England [4,5,6,18]. Interestingly, in 2020, when the initial COVID-19 cases were identified in Rio de Janeiro, Brazil, no cases of IPD were detected in our cohort, probably because cases were missed during this first year of the COVID-19 pandemic.

In a case-control study by Sadlier et al. in PLWHA in Ireland, of 47 episodes of IPD identified in 42 HIV-positive individuals, the incidence of IPD per 100,000 person-years decreased from 728 to 242 to 82 in the periods from 2006–2008, 2009–2012, and 2013–2015, respectively (*p* < 0.01 for linear trend) [4]. In our cohort, of 55 episodes in 45 patients, the average incidence across periods increased from 710 to 1050 to1453 in the calendar periods 2005–2009, 2010–2014, and 2015–2019, respectively, which is different from the study by Sadlier, suggesting IPD is still uniformly high in our scenario. Late diagnosis of HIV, which is reported in a quarter of newly diagnosed HIV patients in Brazil, and low rates of pneumococcus vaccination may have resulted in increased rates of pneumococcus infection. On the other hand, it is possible that implementing a sepsis bundle in our emergency department in 2013 [19,20], leading to more blood culture collection in septic patients, may have influenced our results.

### 4.1. Risk Factors for IPD

Some comorbidities and lifestyle habits that have been described as associated with IPD are male sex [4], smoking [8,9,21], alcoholism [9], injection or inhalation drug use (cocaine, crack) [8,10,21], COPD [8,21] and liver cirrhosis. In our study, COPD and liver cirrhosis were associated with IPD in univariate analysis and were present in almost a quarter of patients. In the adjusted model, there was no statistically significant association. We found no association of IPD with diabetes, corticosteroid use, cancer, and immunosuppressive therapy as described in the literature for non-seropositive patients [22]—this is possibly due to our small sample size and to the fact that our PLWHA are younger and have these conditions less often.

From 2008 to 2021, there was a prevalence of hepatitis B/HIV co-infection of 4.9% and hepatitis C/HIV of 8.3%, according to the 2022 Epidemiological Bulletin, with no identification of the percentage of liver cirrhosis due to these viruses [23].

Low family income [24], education up to high school or low schooling [8], and advanced age > 65 years [4,6,8] were also associated with IPD in some studies. Other studies [5,24] found black skin color to be a risk factor for IPD. Over three-fourth so four patients self-declared as brown or black, and the frequency was similar for both groups. Most patients in both groups had few years of schooling and had low family income.

### 4.2. CD4, ART, and IPD

Recent HIV diagnosis was less frequent among cases. Pneumococcal infections occur in PLWHA even without advanced immunosuppression [25], given the virulence of the pathogen [26]. Several studies show that even with ART use, IPD remains a high-incidence condition in PLWHA [4,6]. In our study, 78.4% of the groups were on ART, with the frequency of use being higher in cases than in controls.

In our cohort, median CD4 counts were low for both cases and controls: 267.5 cells/mm^3^ in cases and 140 cells/mm^3^ in controls. Late presentation to care in PLWHA is common in Latin America [25]. Our controls were patients recently diagnosed with HIV who presented with severe opportunistic diseases, mainly involving the central nervous system (neurotoxoplasmosis and cryptococcosis) and tuberculosis [27]; they were not on ART and had low CD4 counts and high viral load at the time of admission. In other studies, CD4 counts lower than 500 cells/mm^3^ [19], CD4 < 200 cells [8], and CD4 < 100 cells/mm^3^ [6,9] were risk factors for IPD. In addition, having a detectable viral load [4,8] is a risk factor for IPD, also associated with mortality. Munier et al. in 2014 in France (2000–2011), who included 42 patients with IPD and 84 controls (patients without IPD), identified that uncontrolled HIV replication and low CD4 cell counts were risk factors for having IPD [28]. We could not show differences in CD4 counts and viral load between cases and controls, possibly due to our small sample size and our selection of controls, which resulted in a large number of severely ill, hospitalized patients.

### 4.3. Clinical and Laboratory Abnormalities and IPD

Our patients with IPD had significantly more hypotension, tachycardia, tachypnea, fever, and hypoxemia than controls, in agreement with expected responses to sepsis [19]. Most patients hospitalized for IPD had systemic inflammatory response syndrome and, also, met the criteria for infection with organ dysfunction.

Our results are comparable to laboratory results in sepsis, with significantly higher leukocytosis, neutrophilia, band forms, and lymphopenia [19,20], as well as C-reactive protein levels > 5 µg/mL [19,20,29]. As for biochemistry, a higher level of creatinine and total bilirubin was noted in cases that meet the SOFA (Sequential Sepsis-related Organ Failure Assessment) criteria of organ dysfunction. Hypotension, tachycardia, tachypnea, elevated CRP, thrombocytopenia, leukocytosis, band forms, urea, and bilirubin remained significantly different in multivariate analysis between cases and controls.

### 4.4. Mortality and IPD

In-hospital mortality (within 30 days) in our cohort was 21.1% in cases and 14.9% in controls, while mortality in the follow-up period was 42.2% in cases and 40.7% in controls, with no statistical difference between groups.

Although the AIDS mortality rate showed a 17.1% drop in the last 5years in Brazil, there are no exact data for IPD deaths. IPD has been a notifiable condition in Brazil since 2014 [16], and mortality is still unacceptably high [25].

All patients with IPD who died were transferred to the ICU due to sepsis or septic shock. In-hospital mortality in our IPD cases was mainly due to sepsis and was relatively low compared with mortality from sepsis in other studies. Mortality from sepsis in ICU patients in Brazil is around 55% [30]. The mortality due to sepsis in PLWHA in the ICU resulted mainly from lower respiratory tract infections and was about 55% in 2010 [31]; it has decreased over the years, with ICU mortality rates of 32.3% and in-hospital mortality rate of 40.4% in PLWHA in a recent Brazilian study [32].

### 4.5. Antimicrobial Sensitivity

The rate of penicillin resistance was 9%, and penicillin resistance did not correlate with outcomes in our sample. In Brazil, the emergence of pneumococci not susceptible to penicillin has been listed as one of the main threats to antimicrobial-resistant pathogens [33].

Grau et al., in 2005, studying 142 episodes of IPD in 122 HIV patients in Spain, obtained an overall prevalence of strains not susceptible to penicillin of 40% [9], a rate much higher than we found.

Most patients (60%) with IPD were on cotrimoxazole for pneumocystis and toxoplasmosis prophylaxis. Although cotrimoxazole may decrease bacterial infections, antibiotic prophylaxis is not recommended for pneumococcal infection because of the risk of developing resistance [34]. Meynard et al. observed that patients with IPD and on cotrimoxazole prophylaxis had a higher proportion of penicillin-resistant pneumococci [34]. We had a 13% overall pneumococcal resistance to cotrimoxazole.

### 4.6. Pneumococcal Vaccine and HIV

PLWHA are at risk for pneumococcal infections; therefore, pneumococcal vaccination is internationally recommended. Unfortunately, vaccination with PPV23 was documented in only 20% of cases and 18.5% of controls, which is an unacceptably low rate, as we are a referral service with an on-site vaccination clinic and available vaccines. The PCV13 was only made available in the Brazilian public health system in late 2019, and unfortunately, our patients did not benefit from it.

In a Brazilian case-control study by Veras et al. in HIV patients evaluating the efficacy of vaccination on IPD, an efficacy of 63% (95% CI:28–81%) was demonstrated [35].

A blinded clinical trial conducted in Brazil with 331 HIV-positive patients aged 18 to 60 years concluded that both PPV23 and PCV7 demonstrated persistent immunogenicity [36].

One of the limitations of our study is its retrospective nature, which limits the available information found in patients’ notes. Our selection criterion for cases was the presence of IPD, and most of these patients were hospitalized. Therefore, our controls, which were chosen based on age, sex, and scenario of care, were mostly hospitalized. PLWHA who are hospitalized are usually very ill, and this may have influenced our results. Opportunistic diseases in HIV patients, such as cryptococcal meningitis, neurotoxoplasmosis, and other opportunistic diseases, are associated with varying degrees of immunosuppression that increase morbidity and mortality with worse outcomes, and this could also be considered a limitation of our controls. This is a limitation of the study as, naturally, this group of patients will have worse outcomes. A possible limitation is the number of patients included, as IPD is not a commonly identified condition. We must also emphasize our results may not be generalizable, as our practice is in the specific scenario of a research institute in a middle-income country.

## 5. Conclusions

Although ART has been available for all PLWHA since 1996 in Brazil, the pneumococcal disease still has a high incidence in this group. Through a case-control study, we found that patients with HIV and IPD had a higher median CD4 count compared with the control group (267 cells/mm^3^ vs. 140 cells/mm^3^), although both groups could be considered immunosuppressed. In-hospital mortality resulting from sepsis was high in IPDs. Alcoholism, hepatic cirrhosis, and lower nadir CD4 count were associated with the risk of death in patients with IPD. Alcoholism needs to be addressed as a public health hazard. The vaccination rate was less than 20%, which leads us to reinforce the recommendation of vaccination with PCV13 followed by PPV23. Nadir CD4 counts were lower in PLWHA and IPD who died, reinforcing the need for us to diagnose HIV infection early, to start treatment as soon as possible, and to promote adherence to therapy and vaccination.

## Figures and Tables

**Figure 1 tropicalmed-08-00328-f001:**
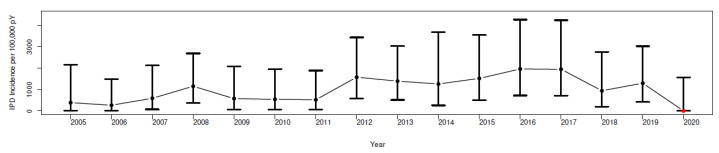
The annual incidence of IPD in PLWHA/100,000 person-years in Rio de Janeiro, Brazil, in 2005–2020.

**Table 1 tropicalmed-08-00328-t001:** Selected demographic and clinical variables of PLWHA on their first episode of IPD compared with PLWHA without IPD.

Variables		All Patients (*n* = 153)	IPD (*n* = 45)	No IPD (*n* = 108)	*p*-Value
		Median [IQR] or *n* (%)	Median [IQR] or *n* (%)	Median [IQR] or *n* (%)	
Age (years)		42 [35;48]	41 [35;47]	42 [35;48]	0.862
Sex at birth	Male	99 (64.7%)	28 (62.2%)	71 (65.7%)	0.819
	Female	54 (35.3%)	17 (37.8%)	37 (34.3%)	
Race/color	Black	41 (26.8%)	13(28.9%)	28 (25.9%)	0.891
	Brown	64 (41.8%)	19 (42.2%)	45 (41.7%)	
	White	48 (31.4%)	13 (28.9%)	35 (32.4%)	
Education	Primary education	58 (39.5%)	15 (35.7%)	43 (41%)	0.689
	Secondary and higher education	89 (60.5%)	27 (64.3%)	62 (59%)	
Smoking	Yes	61 (41.8%)	22 (48.9%)	39 (38.6%)	0.327
Drinking	Yes	47 (32%)	16 (35.6%)	31 (30.4%)	0.67
Drug use	Cocaine/Crack	33 (22.9%)	15 (33.3%)	18 (18.2%)	0.107
	Cannabis	8 (5.6%)	3 (6.7%)	5 (5.1%)	
COPD	Yes	24 (15.8%)	13 (28.9%)	11 (10.3%)	0.009
DM	Yes	8 (5.3%)	4 (8.9%)	4 (3.7%)	0.368
Liver cirrhosis	Yes	15 (9.8%)	9 (20%)	6 (5.6%)	0.015
Steroid use	Yes	8 (5.2%)	3 (6.7%)	5 (4.6%)	0.907
Monthly family income	Up to 3 mw	138 (90.2%)	41 (91.1%)	97 (89.8%)	1
	>3 mw	15 (9.8%)	4 (8.9%)	11 (10.2%)	
Newly diagnosed HIV		28(18.3%)	4(8.9%)	24 (22.2%)	0.087
Nadir(cells/mm^3^)	Nadir	110 [28;275]	117.5 [49.3;316.8]	110 [27;272]	0.606
CD4 (cells/mm^3^)	CD4	165 [53.75;418.25]	267.5 [96.5;495.25]	140 [34;376.75]	0.027
CD4 strata (cells/mm^3^)	<100	54 (37%)	12 (27.3%)	42 (41.2%)	0.025
	100–200	27 (18.5%)	5 (11.4%)	22 (21.6%)	
	>200	64 (44.5%)	27 (61.4%)	38 (37.3%)	
CD4/CD8	CD4/CD8	0.32 [0.09;0.59]	0.4 [0.14;0.68]	0.28 [0.08;0.58]	0.36
Viral load (copies/mL)		1497 [39;79403.75]	793 [39;24821]	2204 [39;158616]	0.311
ART	Yes	120 (78.4%)	40 (88.9%)	80 (74.1%)	0.07
Prophylaxis for OI	Yes	69 (45.1%)	27 (60%)	42 (38.9%)	0.027
PPV23	Yes	29 (19%)	9 (20%)	20 (18.5%)	1
PCV13	Yes	1 (0.7%)	1 (2.2%)	0 (0%)	0.65

PLWHA = people living with the human immunodeficiency virus/acquired immunodeficiency syndrome; IPD = invasive pneumococcal disease; IQR = interquartile range; COPD = chronic obstructive pulmonary disease; DM = diabetes mellitus; mw = minimum wage (Brazil’s minimum wage, as per the government norms, was around US $200.00 per month for the years in the study period). HIV = human immunodeficiency virus; ART = antiretroviral treatment; OI = opportunistic infection; PPV23 = 23-valent pneumococcal vaccine; PCV13 = 13-valent pneumococcal vaccine; *p*-values were calculated for absolute (relative) frequencies by chi-squared tests.

**Table 2 tropicalmed-08-00328-t002:** Physical signs and laboratory features in PLWHA with and without IPD.

Physical Signs and Laboratory Features	All Groups (*n* = 163)	IPD (*n* = 55)	No IPD (*n* = 108)	*p*-Value
	Median [IQR]	Median [IQR]	Median [IQR]	
Systolic blood pressure (mmHg)	110 [100;120]	100 [90;110]	110 [100;120]	<0.001
Diastolic blood pressure (mmHg)	60 [60;80]	60 [60;70]	70 [60;80]	0.002
MAP (mmHg)	80 [73.33;90]	73.33 [66.67;83.33]	83.33 [73.33;93.33]	<0.001
HR	101 [86;120]	112 [93;124.5]	97 [82;114]	0.014
RR	20.5 [19;27]	25 [20.5;28.5]	20 [18;24]	<0.001
Temperature	37 [36.1;38.08]	37.65 [36.53;38.88]	36.8 [36;38]	0.019
Room SPO_2_	97 [92.75;98]	95 [90;97]	98 [94.5;99]	0.008
BMI (kg/m^2^)	20.3 [17.95;23.7]	21.3 [18.6;24.9]	19.6 [17.93;23.53]	0.349
Hemoglobinlevels (g/dL)	10.8 [9;12.9]	10.75 [9.15;12]	10.9 [8.95;13.35]	0.447
Platelets (10³/mm³)	224 [157;293]	214 [151.75;297.5]	232 [168.5;284.5]	0.791
Leukocytes (10^3^/mm^3^)	7300 [4645;12,535]	12675 [8680;15,395]	6180 [4055;8550]	<0.001
Bands (%)	5 [2;10.75]	7.5 [4.75;15.25]	3 [2;8.75]	<0.001
Segmented Neutrophils (%)	70 [58;77]	72 [63.5;80]	65.5 [55.25;76]	0.017
Lymphocytes (%)	14 [8;24]	8.5 [5.75;14.25]	17 [11;30]	<0.001
Creatinine (mg/dL)	1.04 [0.8;1.64]	1.64 [1.08;2.55]	0.96 [0.76;1.29]	<0.001
Urea (mg/dL)	33 [25;54]	51.5 [30;96.5]	32 [22;43]	<0.001
AST (IU/L)	34 [25;61.25]	48.2 [25;70.75]	33 [25;50]	0.07
ALT (IU/L)	31.8 [25;50.75]	30.5 [24.25;46.75]	33.5 [25;52.25]	0.408
Total Bilirubin (mg/dL)	0.49 [0.28;0.86]	0.57 [0.34;1.26]	0.44 [0.27;0.8]	0.074
Albumin (g/L)	2.3 [1,9;2,98]	2.1 [1,8;2.6]	2.6 [2;3.1]	0.013
Sodium (mEq/L)	134 [130;138]	133 [130;136.5]	135 [132;138]	0.061
CRP levels (mg/dL)	9.2 [3.14;20.33]	22 [10.45;28.67]	5.77 [1.14;11.51]	<0.001

PLWHA = people living with the human immunodeficiency virus/acquired immunodeficiency syndrome; IPD = invasive pneumococcal disease; IQR = interquartile range; MAP = mean arterial pressure; HR = heart rate; RR = respiratory rate; SPO_2_ = peripheral blood oxygen saturation; BMI = body mass index; AST = aspartate aminotransferase; ALT = alanine aminotransferase; CRP = C-reactive protein; *p*-values were calculated for the median [interquartile range] by Mann–Whitney (Wilcoxon rank-sum test).

**Table 3 tropicalmed-08-00328-t003:** Analysis of adjusted factors associated with having an infection (IPD) among HIV+ individuals, taking into account the first episode.

Variables	Level	IPD		OR (CI95%)	*p*-Value	aOR (CI95%)	*p*-Value
		Yes (*n* = 45)	No (*n* = 108)				
BMI (kg/m^2^)	>18	6 (28.57%)	22 (28.95%)	Ref.	Ref.	Ref.	Ref.
	19–25	12 (57.14%)	42 (55.26%)	1.05 (0.35–3.17)	1	1.99 (0.38–10.3)	0.828
	>26	3 (14.29%)	12 (15.79%)	0.92 (0.19–4.34)	1	1.59 (0.15–16.57)	0.828
MAP (mmHg)	>70	26 (61.9%)	87 (84.47%)	0.3 (0.13–0.68)	0.003	0.17 (0.05–0.53)	0.002
HR	>90	32 (78.05%)	60 (60.61%)	2.31 (1–5.37)	0.051	2.51 (0.78–8.1)	0.123
RR	>22	28 (68.29%)	30 (31.58%)	4.67 (2.12–10.25)	0.000	4.79 (1.5–15.29)	0.008
Hemoglobin (g/dL)	>10	27 (61.36%)	67 (63.21%)	0.92 (0.45–1.91)	0.831	1 (0.3–3.28)	0.998
CRP (mg/dL)	>10	17 (73.91%)	21 (31.34%)	6.21 (2.14–17.99)	0.000	9.16 (1.98–42.38)	0.004
Platelets (10³/mm³)	>225	19 (43.18%)	55 (51.89%)	0.7 (0.35–1.43)	0.332	0.24 (0.08–0.72)	0.011
Leukocytes (10^3^/mm^3^)	>7300	33 (76.74%)	37 (34.91%)	6.15 (2.73–13.87)	<0.000	3.45 (1.12–10.56)	0.030
Bands (%)	>6	30 (71.43%)	36 (34.29%)	4.79 (2.19–10.47)	<0.000	4.14 (1.36–12.63)	0.012
Segmented Neutrophils (%)	>70	21 (50%)	43 (40.95%)	1.44 (0.7–2.96)	0.318	3.24 (1.07–9.76)	0.037
Lymphocytes (%)	>15	12 (28.57%)	64 (60.95%)	0.26 (0.12–0.56)	0.000	0.09 (0.02–0.37)	<0.001
Creatinine (mg/dL)	>1	32 (76.19%)	38 (35.85%)	5.73 (2.54–12.92)	0.000	9.52 (2.88–31.45)	<0.001
Urea (mg/dL)	>33	27 (67.5%)	42 (39.62%)	3.16 (1.47–6.82)	0.003	3.83 (1.24–11.81)	0.019
Bilirubin (mg/dL)	>0.50	24 (66.67%)	44 (43.56%)	2.59 (1.17–5.75)	0.019	4.19 (1.23–14.25)	0.021
Albumin (g/L)	>2.3	8 (33.33%)	35 (56.45%)	0.39 (0.14–1.03)	0.058	0.05 (0–0.63)	0.020

PLWHA = people living with the human immunodeficiency virus /acquired immunodeficiency syndrome; IPD = invasive pneumococcal disease; HIV = human immunodeficiency virus; MAP = mean arterial pressure; HR = heart rate; RR = respiratory rate; SPO_2_ = peripheral blood oxygen saturation; BMI = body mass index; CRP = C-reactive protein; OR (CI95%) = odds-ratio of having an infection (IPD) among levels of the features related to the reference and its 95% confidence intervals estimated by Logistic (Binomial) regression models. aOR (CI95%) = ORs adjusted for confounding variables (i.e., patients’ age, birth sex, race/color, level of education, household income, nadir T-CD4+ count, T-CD4+ count, and viral load (the latter two being measured in the last six months of follow-up) and its 95% confidence intervals estimated by multiple Logistic (Binomial) regression models.

**Table 4 tropicalmed-08-00328-t004:** Adjusted risk factors associated with IPD among PLWHA and IPD, taking into account the first episode.

Variables	Level	IPD		OR (CI95%)	*p*-Value	aOR (CI95%)	*p*-Value
		Yes (*n* = 45)	No (*n* = 108)				
Sex at birth	Female	17 (37.78%)	37 (34.26%)	1.17 (0.57–2.4)	0.678	1.55 (0.56–4.31)	0.397
Smoking	Yes	22 (48.89%)	39 (38.61%)	1.52 (0.75–3.09)	0.246	1.84 (0.65–5.18)	0.250
Drinking	Yes	16 (35.56%)	31 (30.39%)	1.26 (0.6–2.65)	0.536	1.89 (0.68–5.29)	0.223
Drug use	Cocaine/crack	15 (33.33%)	18 (18.18%)	2.35 (1.04–5.29)	0.080	1.5 (0.49–4.58)	0.956
	Cannabis	3 (6.67%)	5 (5.05%)	1.69 (0.38–7.55)	0.492	NC	NC
COPD	Yes	13 (28.89%)	11 (10.28%)	3.55 (1.45–8.7)	0.005	2.88 (0.79–10.54)	0.109
DM	Yes	4 (8.89%)	4 (3.74%)	2.51 (0.6–10.52)	0.207	2.38 (0.31–18.11)	0.403
Liver cirrhosis	Yes	9 (20%)	6 (5.56%)	4.25 (1.41–12.78)	0.009	3.58 (0.7–18.22)	0.125
Steroid use	Yes	3 (6.67%)	5 (4.63%)	1.47 (0.34–6.44)	0.607	0.39 (0.04–4.12)	0.430
New case HIV	Yes	4 (8.89%)	24 (22.22%)	0.34 (0.11–1.05)	0.060	NC	NC
ART Use	No	5 (11.11%)	28 (25.93%)	0.36 (0.13–0.99)	0.049	0.66 (0.11–3.9)	0.644
CD4 (cells/mm^3^)	<100	12 (27.27%)	42 (41.18%)	Ref.	Ref.	Ref.	Ref.
	100–200	5 (11.36%)	22 (21.57%)	0.8 (0.25–2.55)	0.699	0.41 (0.07–2.42)	0.655
	>200	27 (61.36%)	38 (37.25%)	2.49 (1.11–5.59)	0.055	1.46 (0.24–8.95)	0.683
Prophylaxis for OI	Yes	27 (60%)	42 (38.89%)	2.36 (1.16–4.8)	0.018	2.86 (0.87–9.36)	0.082

PLWHA = people living with HIV/AIDS; IPD = invasive pneumococcal disease; HIV = human immunodeficiency virus; COPD = chronic obstructive pulmonary disease; DM = diabetes mellitus; ART = antiretroviral treatment; OI = opportunistic infection; OR (CI95%) = odds-ratio of having an infection (IPD) among levels of the features related to the reference and its 95% confidence intervals estimated by Logistic (Binomial) regression models; aOR (CI95%) = ORs adjusted for confounding variables (i.e., patients’ age, birth sex, race/color, level of education, household income, nadir T-CD4+ count, T-CD4+ count, and viral load (the latter two being measured in the last six months of follow-up) and its 95% confidence intervals estimated by multiple Logistic (Binomial) regression models.

## Data Availability

Research data has not been made unavailable due to privacy or ethical restrictions.

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
