# Peer review of "Invasive Pneumococcal Disease in People Living with HIV: A Retrospective Case—Control Study in Brazil"

_tropicalmed, 2023, doi:10.3390/tropicalmed8060328_

Round 1

Reviewer 1 Report

Matherial and metrod - please describe the method of isolation of Streptococcus pneumoniae from pathological products

Results - there are no data on antibiotic resistance (antibiograms)

Table 2. Physical signs and laboratory features in PLWHA with and without IPD. - I don't think this information is relevant. Obviously, patients with HIV and pneumococcal infections had fever, tachycardia, leukocytosis, neutrophilia, etc., compared to the control group

it is difficult to compare the mortality in the group of patients with HIV and pneumococcal infections and the control group, it is not mentioned what are the causes of death in the control group

Matherial and metrod - please describe the method of isolation of Streptococcus pneumoniae from pathological products

Results - there are no data on antibiotic resistance (antibiograms)

Table 2. Physical signs and laboratory features in PLWHA with and without IPD. - I don't think this information is relevant. Obviously, patients with HIV and pneumococcal infections had fever, tachycardia, leukocytosis, neutrophilia, etc., compared to the control group

it is difficult to compare the mortality in the group of patients with HIV and pneumococcal infections and the control group, it is not mentioned what are the causes of death in the control group

Author Response

We thank the reviewer for his/her comments, which were very helpful.

Comments and Suggestions for Authors

  1. Material and method - please describe the method of isolation of Streptococcus pneumoniae from pathological products

R: We have added the following description in the methods section:

“The isolation method was performed by direct inoculation of clinical samples in enriched culture media (chocolate agar, blood agar) and incubation at 5% CO2, 35 +/-1°C, within 72 hours. Blood and/or sterile fluid samples were inoculated into automated culture flasks (Bactec/Becton Dickson). Incubation up to 5 days. The antimicrobial susceptibility test was performed by the agar diffusion method (disk diffusion) and determination of the minimum inhibitory concentration (gradient strip agar diffusion). Following the interpretative criteria of the CLSI- Clinical and Laboratory Standards Institute. Molecular tests and serotyping were not performed”.

  1. Results - there are no data on antibiotic resistance (antibiograms)

R: We have provided the sensitivity in the results section:

  1. Table 2. Physical signs and laboratory features in PLWHA with and without IPD. - I don't think this information is relevant. Obviously, patients with HIV and pneumococcal infections had fever, tachycardia, leukocytosis, neutrophilia, etc., compared to the control group

R: We would like to keep the physical signs and laboratory features, as we think this better describes the patients affected by invasive pneumococcal disease.

  1. it is difficult to compare the mortality in the group of patients with HIV and pneumococcal infections and the control group, it is not mentioned what are the causes of death in the control group.

R: We have given the causes of death in the results section, as below:

We had 15 patients in the control group who died (14.9%), in 8/15 (53%) the  cause was respiratory (pulmonary tuberculosis, pneumocystis pneumonia, other bacterial pneumonia), in  5/15 (33%), the cause was neurological (meningoencephalitis, neurotoxoplasmosis, AIDS encephalitis), in 2/15 (13%) the cause was hematological (Kaposi’s sarcoma and lymphoma).

Author Response

Brief summary:

This is a retrospective case-control study looking at risk factors associated with invasive pneumococcal disease among people living with HIV/AIDS. The study recruited cases and controls from an HIV clinic in Brazil over a 16-year period. Cases included those with laboratory-confirmed IPD and controls were matched by age, sex, semester of IPD infection and setting (whether IPD was diagnosed as an outpatient or in the emergency department, or whether the patient with IPD was hospitalised or in ICU). Incidence of IPD amongst PLWHA attending the HIV clinic increased over the study period, with a case-fatality of 21% amongst those with IPD. Liver cirrhosis was the only significant risk factor for developing IPD amongst PLWHA. Clinically, PLWHA hospitalised with IPD were more likely to have lower mean arterial pressure, higher respiratory rate, higher CRP, lower platelets, lower leukocytes, higher bands, higher segmented neutrophils, lower lymphocytes, higher urea, higher bilirubin and lower albumin than matched hospitalised controls. PLWHA with IPD were no more likely to die from the IPD episode or within one year of the episode than matched controls. This is a comprehensive analysis of IPD in PLWHA in the context of Brazil. It is a well written, methodical and detailed analysis but would do even better if it could have been contextualised more with regards to the situation of IPD in the general population.

General concepts:

Case control studies are only as good as the control selection, as controls need to represent the source population from which the cases are drawn. Retrospective studies make this a little more difficult and the authors need to be aware that the incorrect selection of controls could introduce bias.

R: When we set out to do this study, we thought controlling by age, gender, scenario of care and semester/year would be the most simple and honest way to match our cases. We were surprised by how sick our controls turned out to be, making patients with IPD seem a better group of patients, despite the potential severity of IPD with resulting sepsis and associated death.

I have mentioned below a number of areas in which the study could be strengthened or adjusted.

R: We thank you for your very thorough review. By answering to each question raised, we hope to have improved our manuscript. Specific comments:

Please make sure you define all abbreviations/acronyms before using them (especially in the abstract). ART, CI, OR, Pn23 etc.

R: We have reviewed all abbreviations and acronyms and defined them, including in the abstract.

Abstract:

Line 35: “Bacterial pneumonia” – this is a confusing term when referring to the IPD group. Please define this better in the methods too. Bacterial pneumonia refers to all pneumonias of bacterial origin (not just pneumococcus), and it is not confined to invasive disease (sputum positive could also be defined as bacterial pneumonia).  Rather say “76.4% of IPD episodes presented with pneumonia and 20% with bacteraemia without focus.

R: We agree and have rewritten the results as suggested.

Line 43,44: “Low CD4 counts were associated with death” …this was not given as a result in the abstract, therefore you can’t introduce it in the conclusion as a main finding.

R: We have shown, in Supplementary Table 5, that the median nadir CD4 count was lower in the patients with IPD who died, compared to those who lived. We have added this result to the written results section. We have highlighted in bold and yellow, in Supplementary Table 5, all statistically significant differences between those with IPD who died vs those who lived. Because this variable was not included in the adjusted model, we removed it from the conclusions in the abstract.

Introduction:

Line 55 and 56: please provide references

R: These have been provided:

Feikin DR, Feldman C, Schuchat A, Janoff EN. Global strategies to prevent bacterial pneumonia in adults with HIV disease. Lancet Infect Dis. julho de 2004;4(7):445–55.

Lines 64-70: consider providing more up to date references ie Kirwan, 2021, CID; etc

R: We have included the mentioned reference.

Kirwan PD, Amin-Chowdhury Z, Croxford SE, Sheppard C, Fry N, Delpech VC, et al. Invasive Pneumococcal Disease in People With Human Immunodeficiency Virus in England, 1999-2017. Clin Infect Dis Off Publ Infect Dis Soc Am. 1o de julho de 2021;73(1):91–100.

I am not convinced of the need for this particular study to be published in an international journal. Please contextualise the study setting and motivate how this descriptive analysis may add to and assist in the global understanding of IPD and HIV.

R: Invasive pneumococcal disease affects all human beings, irrespective of geographical location. Brazil is a middle-income country, located in South America. The Brazilian scenario, therefore, may apply to other Latin American countries and other middle-income countries. We did not find literature from low and middle-income countries. These are the reasons our study merits publication in an international journal and contributes to the global understanding of IPD and HIV.

Methods:

Lines 89-92: Study population…were the controls matched by year as well as semester?

R: By matching by semester, we mean the same semester and year.

Line 90 and 93: Gender and sex at birth? Are these different variables? If not, please stick to one variable name? Correct this in the tables too.

R: Gender and sex at birth are the same variable, we will keep the term sex at birth

Control selection is extremely important and incorrect selection of controls can introduce bias. Were all inpatient controls medical admissions or were some surgical admissions and do you think this difference could influence your results. Regarding the emergency room matching – were these patients seen in ER and then discharged or were some admitted thus straddling the two categories for matching? Please describe your rationale for selection of matching criteria of controls.

R: All inpatient controls were medical admissions, as we are a medical unit, having no onsite surgical facilities.  All patients who were seen in ER and then discharged remained categorized to this setting; if they were admitted, they were categorized as admitted. Therefore, no straddling of the two categories for matching occurred.

Lines 106-107: Community acquired infection …if event occurred within 48hrs of admission? What if the IPD was diagnosed as an outpatient.

R: We have made the correction as below:

“Community-acquired infection was considered if the patients remained outpatients and had no previous recent hospital admissions, or if admitted, the diagnosis occurred up to 48 hours after admission , and hospital-acquired infection if the event occurred more than 48 hours after admission”

Line 108: see comment above about “bacterial pneumonia” … perhaps add into your definition here that it is pneumococcus cultured from a sterile specimen with symptoms of pneumonia. “Bacteraemic pneumonia” is probably the term that you should have used in the abstract.

R: We have added the need for isolation from a sterile sample in pneumonia, as follows: “Pneumonia was defined as an acute or subacute pulmonary condition characterized by: fever and/or cough; new purulent sputum; dyspnea or tachypnea and pleuritic chest pain, with or without radiological confirmation, and with adequate antibiotic treatment administered [10] as well as isolation of pneumococcus in a sterile sample such as blood or pleural fluid;

Line 118: specify which vaccine or any pneumococcal vaccine?

R: We mean any pneumococcal vaccine.

Line 137 and 138: if you are matching cases and controls by age and sex you need not include them as confounders in your models.

R: Please find below our statistician’s detailed reply:

Thank you for the careful review, questioning, and the opportunity to clarify the issue. We understand the reviewer's point and agree almost entirely with his statement. In fact, in a case-control study, where controls are paired by some characteristic, in this case, sex and age, there would be no need to include these characteristics in association analysis of any other characteristic between cases and controls, traditionally performed via generalized linear models with logistic linkage family and association measure presented as Odds-ratios. However, already in the data analysis phase, after reviewing and tabulating the data of cases and their matched controls, we decided on association analyses between the first cases of IPD (i.e., for patients who had IPD recurrence, we included only their first case) and controls. Because of this decision, we reduced from 55 cases that included relapses to 45 independent cases (i.e., from different patients) and 108 controls; remembering that we intended to select two age- and sex-matched controls with each case. Thus, we no longer had an exact match between cases and controls after categorizing the age variable (using the sample median as a cutoff point). We now had 48 and 51 males and 24 and 30 females in the age categories of (17,40) and (40,65), respectively, close to the original intent. In addition, we also performed time-to-event analyses in the case of death (or hospital discharge) after an IPD episode, an analysis that employed multiple Cox proportional hazards models and included only IPD cases. In the latter, we had an even more significant imbalance between men and women in the same age groups. More precisely, we had 16 and 12 men and 6 and 11 women in the age categories of (17,40] and (40,65], respectively. We also had an imbalance of the mean (standard deviation) time (in years) of follow-up between men, 3.66 (0.9) and 4.46 (1.46), and women, 5.64 (1.87) and 3.22 (1.11), in the age categories of (17,40] and (40,65], respectively. For the reasons mentioned above, we thought it was the best strategy to include the variables age and sex, among others (e.g., race/color, level of education, household income, Nadir T-CD4+ count, T-CD4+ count, and viral load), in both analyses. With this, we had a single list of confounding variables independent of the analysis, which would facilitate the reading and reproduction of the results by independent research groups. It is a fact that in the simple binomial regression models (bivariate analyses), there was no evidence, or even a trend, of an association between the average incidence of IPD between the levels of the sex variable, with an OR (CI95%) estimated at 1.17 (0. 57-2.4) and p-value of 0.68 between women and men, as well as between the levels of the variable age categorized by sample median, with OR (CI95%) estimated at 0.9 (0.45-1.81) and p-value of 0.77 between individuals aged between (40.65] and (17.40]. A similar result was also observed in the time-to-event analysis between women and men, estimated HR (CI95%) of 0.497 (0.1-2.467), a p-value of 0.39, but there was a slight trend (p-value <= 0.2) of more significant progression to death for older individuals, estimated HR (CI95%) of 2.83 (0.567-14.112), a p-value of 0. 2. although in our sample, regardless of the analysis, we found no evidence/trend of higher or lower risk for women (relative to men) we decided to keep it in the multiple regression models understanding that its estimated parameter is very close to zero and therefore negligible.

Line 148: Please refrain from ranking levels of significance based on p-values, the p-value either indicates that it is significant or not.

R: We agree with you, and we have excluded terms such as "highly significant", "significant" and "suggestive".

Lines 152: I am unsure of the journal policies but if possible please mention here whether or not consent was obtained from the patients and give the ethics committee reference number.

R: The journal usually requires the Ethics Committee statement at the end, just before the references. But we have added to the Methods the following sentences: “This is a retrospective and descriptive study and the participants ' consent was not deemed necessary by   the local Ethics Committee.The study was approved by the Ethics Committee of the Instituto Nacional de Infectologia Evandro Chagas/ Fiocruz (IRB n° 32449420.4.1001.5262), number 4,133,994 on 13 July 2020.

Results:

Line 160-162: It is not essential but would be good to see the percentage increase and significance of the increasing trend overtime.

R: We thank the reviewer for his careful revision and valuable suggestion. We have decided to accept the suggestion and improve the paragraph's wording to clarify the interpretation of the results, referring to Figure 2. Where there was "In the calendar periods 2005-2009, 2010-2014, and 2015-2019, the incidence of IPD per 100,000 person-years increased from 710 (95% CI, 270-1150), to 1050 (95% CI, 520-1580), to 1453 (95% CI, 946-1960), respectively., we added "We found no significant differences between the years 2005 and 2019 for the mean incidences of IPD per 100,000 person-years, as evidenced by the overlap of the estimated 95% confidence intervals, but we did find a trend of a 1.4% increase per year (rho=0.014; p-value < 0.001)." Also, we included the following in the methodological description of this latter analysis in the Statistical Analysis section "To identify an increasing or decreasing trend in IPD, we used non-parametric Spearman test between the observations and time estimated in bootstrap samples (R=1000)."

Line 163: I am unsure why you switch between using episodes (n=55) and patients (n=45) in the analysis? Surely the patients who had recurrent episodes could have different risk factors over the time period of their infections and it would be worth reporting on characteristics of the episode rather than that of the patient (which could have changed from the first episode)? See all tables.

R: Previous reviewers have asked us to analyse results as we have, as episodes and patients. Comorbidities and sex do not change between episodes, while habits, clinical-laboratory features and outcomes may be different.

Line 168: “The majority of patients were single” you do not mention collecting data on marital status in the methods, is this necessary to report here?

R: Marital status does not seem important regarding invasive pneumococcal disease, and we therefore decided to exclude this result.

Line 204: perhaps mention your main findings of this table in this paragraph as you include it in your conclusion of your abstract.

R: We have added these results, as suggested, as follows: “Adjusted demographic and clinical characteristics of PLWHA with IPD who died in hospital are presented in Supplementary Table 1, only alcoholism (p <0.018) and hepatic cirrhosis (p <0.003) remained associated with the risk of death in patients with IPD. “

The high numbers of controls with neurological syndromes may have influenced your results. In particular, cryptococcal meningitis typically occurs in patients with exceptionally low CD4 counts and they have a terrible prognosis even with great treatment. I would suggest that you discuss this later in the discussion section as a limitation in your study.

R: We have done so, by including the following sentences to the discussion’s limitations of our study.

“Opportunistic diseases in HIV patients, such as cryptococcal meningitis, neurotoxoplasmosis, and other opportunistic diseases are associated with varying degrees of immuno-suppression that increase morbidity and mortality with worse outcomes, and this could also be considered a limitation of our controls. Opportunistic diseases in HIV patients, such as cryptococcal meningitis, neurotoxoplasmosis, and other opportunistic diseases are associated with varying degrees of immunosuppression that increase morbidity and mortality with worse outcomes, this could also be considered a limitation.”

Table 2 and 3 could be combined into one table.

R: We would like to keep the tables separate, as a lot of information is given in each one of them.

Regarding the factors associated with IPD infection, since many of these deranged chemistry markers may be driven by other comorbidities would it not make sense to control for them in the multivariable analysis ie liver cirrhosis?

R: We have taken advice from our statistician, and he tells us it was not possible to carry out this analysis, since our sample is small, we have 8 patients with IPD who died, and of these, 5 had liver cirrhosis and 3 did not.

Discussion:

It would help to start your discussion with a summary of your main findings. Ie incidence, mortality, risk factors for IPD and associated attributes during IPD infection. And then use the rest of the discussion to unpack these.

R: We thank you for the suggestion. We have started the discussion with the sentences:

“Ours was a retrospective study on IPD in PLWHA in Rio de Janeiro, Brazil, in the years 2005 to 2020. The incidence of IPD was high as was the associated in-hospital mortality in patients living with HIV in our cohort during the study period.  Alcohol abuse, liver cirrhosis and COPD were the only factors associated with IPD in PLWHA, and alcoholism and liver cirrhosis were the only factors associated with death”.

Lines 272: Give a background of HIV prevalence in Brazil over the years…has it gone up or down or remained steady, to explain the rise in incidence. Also it would be worth noting the overall population incidence of pneumococcal disease in Brazil over a similar time period – has it also increased? Are children immunised against IPD – if so, when was this introduced, what is the coverage and what is the potential for herd immunity?

R: We thank you for these suggestions. We have added the following information ,with the references related to them , to the Discussion:

In Brazil, from 1980 to June 2022, HIV prevalence decreased by 26.5%, from 22.5 cases/100,000 inhabitants in 2011 to 16.5 cases/100,000 inhabitants in 2021, according to the Bulletin epidemiological data for 2022.Although there has been a report on the decrease in AIDS cases in recent years, it should be noted that part of this reduction may be related to underreporting of cases, especially in 2020, due to the covid-19 pandemic (Ministerio de Saude M de S. Boletim Epidemiológico - HIV/Aids 2022 — Departamento de HIV/Aids, Tuberculose, Hepatites Virais e Infecções Sexualmente Transmissíveis)

Regarding the prevention of IPD and pneumococcal pneumonia, in Brazil, the 10-valent pneumococcal vaccine has been introduced since 2010, through the PNI (National Immunization Program). Currently, three pneumococcal vaccines are registered for use: Pn10, Pn13 and Pn23, the vaccination scheme for children under 5 years and with Pn10 (2 doses and 1 booster), for adults over 60 years 1 dose of Pn23 , and for patients living with HIV and other immunosuppressed 1 dose of Pn13 followed by Pn23 six to 12 months later, and a second dose of Pn23 five years after the first.

According to data from the Ministry of Health, the vaccination coverage of the population has been decreasing, with lowest percentages found in 2021 were less than than 59% of the target group of patients immunized. In 2020, the index was 67% and in 2019, 73%. (20221220_vacina_pneumococica13_cp_98.pdf Available at: https://www.gov.br/conitec/pt-br/midias/consultas/relatorios/2022/20221220_vacina_pneumococica13_cp_98.pdf)

Line 292: what do you mean by “non-seropositive” patients?

R: We mean that these patients are not HIV positive.

Lines 285: Paragraph discussion comorbidities…what is the prevalence of Hepatitis C or B amongst PLWHA in Brazil? Is this linked with the liver cirrhosis or is alcoholism or other aetiologies. How much of a risk factor for IPD is liver cirrhosis in the non-HIV infected population?

R: We have added the following information to the Discussion: “From 2008 to 2021, there was a prevalence of Hepatitis B/HIV co-infection of 4.9% and Hepatitis C/HIV of 8.3%, according to the 2022 Epidemiological Bulletin, with no identification of the percentage of liver cirrhosis due to this cause, while liver cirrhosis due to other causes has not been reported”. (Boletim-epidemiologico-de-hepatites-virais-2022)

Lines 306 onward: Please elaborate on your selection of controls as a limitation to the study. I suspect that many of them were severely immunocompromised and perhaps not matching the control group to cases through hospitalisation or ICU would have been a better choice. Perhaps matching by timing of HIV diagnosis would have controlled better for those who were slow to seek HIV care, whilst still allowing you to investigate CD4 and viral load as risk factors for IPD.

R: When we set out to do this study, we thought controlling by age, gender, scenario of care and semester/year would be the most simple and honest way to match our cases. We were surprised by how sick our controls turned out to be, making patients with IPD seem a better group of patients, despite the potential severity of IPD with resulting sepsis and associated death.

Lines 321 onward: Include a discussion on the abnormalities particularly related to those with IPD and liver cirrhosis. These may have influenced many of the laboratory results. Also mention the poor inflammatory response in those controls with cryptococcosis in which many may not have had any inflammatory cells in the CSF due to immunosuppression, etc from advanced HIV.

R: It was not possible to include the discussion regarding IPD and liver cirrhosis because of the very small numbers of patients, which precludes a robust statistical analysis. Only 8 patients with IPD died, of which only 5 had liver cirrhosis.

We did not discuss in detail the lack of inflammatory response in the CSF of meningeal cryptococcosis, as we do not find this is a central issue in discussing IPD vs controls. All controls had severe opportunistic infection, such as disseminated tuberculosis, neurotoxoplasmosis, Kaposi sarcoma.

In the final paragraph of your conclusion beware mentioning items that were not discussed earlier or reported in the results. It is best to summarise your main findings and suggested recommendations in light of your findings.

R: We have provided results which were not explicit in the text before. Our conclusions read as follows:

“Although ART has been available for all PLWHA since 1996 in Brazil, pneumococcal disease still has a high incidence in this group. Through a case-control study, we found that patients with HIV and IPD had a higher median CD4 count compared to the control group (267 cells/mm3 vs 140 cells/mm3), although both groups could be considered immunosuppressed. Alcoholism and liver cirrhosis were the only factors associated with IPD in PLWHA, indicating the need to incisively address these comorbidities and habits in patients. In-hospital mortality was high in IPDs, resulting from sepsis. The vaccination rate was less than 20%, which leads us to reinforce the recommendation of vaccination with Pn13V followed by PnPV23. Nadir CD4 counts were lower in PLWHA and IPD who died, reinforcing the need for us to diagnose HIV infection early, to start treatment as soon as possible, and to promote adherence to therapy and vaccination.”

Figures and tables:

Figure 1: Change “DPI” on Y-axis to IPD.  And remove header in the figure (Graph1…)

R: We have done so, thank you.

Table 1: note interquartile range or row/column percentage in header

  1. We have inserted the terms Median[IQR] or n(%) in a line in Table 1, and highlighted it in yellow.

Supplementary Table 2: Not sure what the percentage in brackets is referring to – is it a column percentage? Please update the headers to describe these findings.

R: In table 2, the brackets refer to IQR; we have added a line with the terms Median[IQR] and highlighted it in yellow.

Round 2

Reviewer 1 Report

 Accept in present form 

Author Response

We thank you for all your valuable comments. Our paper is certainly improved after we modified it to include your suggestions.
